# Relationship between IgA Nephropathy and *Porphyromonas gingivalis*; Red Complex of Periodontopathic Bacterial Species

**DOI:** 10.3390/ijms222313022

**Published:** 2021-12-01

**Authors:** Yasuyuki Nagasawa, Ryota Nomura, Taro Misaki, Seigo Ito, Shuhei Naka, Kaoruko Wato, Mieko Okunaka, Maiko Watabe, Katsuya Fushimi, Kenzo Tsuzuki, Michiyo Matsumoto-Nakano, Kazuhiko Nakano

**Affiliations:** 1Department of Internal Medicine, Division of Kidney and Dialysis, Hyogo College of Medicine, Nishinomiya 663-8501, Hyogo, Japan; 2Department of Pediatric Dentistry, Division of Oral infection and Disease Control, Osaka University Graduate School of Dentistry, Suita 565-0871, Osaka, Japan; wato@dent.osaka-u.ac.jp (K.W.); nakano@dent.osaka-u.ac.jp (K.N.); 3Division of Nephrology, Seirei Hamamatsu General Hospital, Shizuoka 430-8558, Hamamatsu, Japan; misakitar@gmail.com; 4Department of Nursing, Faculty of Nursing, Seirei Christopher University, Shizuoka 433-8558, Hamamatsu, Japan; 5Department of Internal medicine, Japan Self-Defense Gifu Hospital, Gifu 502-0817, Kakamigahara, Japan; seigoemon@yahoo.co.jp; 6Department of Pediatric Dentistry, Okayama University Graduate School of Medicine, Dentistry and Pharmaceutical Sciences, Okayama 700-8525, Okayama, Japan; nshuhei@okayama-u.ac.jp (S.N.); mnakano@cc.okayama-u.ac.jp (M.M.-N.); 7Department of Otolaryngology, Meiwa Hospital, Nishinomiya 663-8186, Hyogo, Japan; sone@meiwa-hospital.com (M.O.); maichansan0101@gmail.com (M.W.); fussybilliard@gmail.com (K.F.); 8Department of Otorhinolaryngology, Head and Neck Surgery, Hyogo College of Medicine, Nishinomiya 663-8501, Hyogo, Japan; kenzo@hyo-med.ac.jp

**Keywords:** IgA nephropathy, periodontal bacteria, infection, mouse, tonsil, oral bacteria, *Porphyromonas gingivalis*

## Abstract

IgA nephropathy (IgAN) has been considered to have a relationship with infection in the tonsil, because IgAN patients often manifest macro hematuria just after tonsillitis. In terms of oral-area infection, the red complex of periodontal bacteria (*Porphyromonas gingivalis* (*P. gingivalis*), *Treponema denticol (T. denticola)* and *Tannerella forsythia (T. forsythia)*) is important, but the relationship between these bacteria and IgAN remains unknown. In this study, the prevalence of the red complex of periodontal bacteria in tonsil was compared between IgAN and tonsillitis patients. The pathogenicity of IgAN induced by *P. gingivalis* was confirmed by the mice model treated with this bacterium. The prevalence of *P. gingivalis* and *T. forsythia* in IgAN patients was significantly higher than that in tonsillitis patients (*p* < 0.001 and *p* < 0.05, respectively). A total of 92% of tonsillitis patients were free from red complex bacteria, while only 48% of IgAN patients had any of these bacteria. Nasal administration of *P. gingivalis* in mice caused mesangial proliferation (*p* < 0.05 at days 28a nd 42; *p* < 0.01 at days 14 and 56) and IgA deposition (*p* < 0.001 at day 42 and 56 after administration). Scanning-electron-microscopic observation revealed that a high-density Electron-Dense Deposit was widely distributed in the mesangial region in the mice kidneys treated with *P. gingivalis*. These findings suggest that *P. gingivalis* is involved in the pathogenesis of IgAN.

## 1. Introduction

IgA nephropathy (IgAN) is one of most common primary glomerulitis nephropathies not only in Japan but also in the world [1,2]. The pathogenesis of IgAN was considered to consist of genetic factors [3,4,5,6] and environmental factors [7,8,9], including infection [10]. When IgAN patients suffer from tonsillitis, the hematuria sometimes worsens faster than usual, and the removal of tonsils from the IgAN patients ameliorated proteinuria and hematuria [11,12,13]. These features of IgAN indicated that some part of pathogenesis of IgAN has a relationship with infection in oral areas, including the tonsils. 

In terms of pathogenic infections in oral areas, cariogenic bacteria and periodontal bacteria are important. *Streptococcus mutans* (*S. mutans*) is well-known to have strong cariogenicity; *cnm*-positive *S. mutans* has a stronger collagen-binding capacity than *cnm*-negative *S. mutans*; and *cnm*-positive *S. mutans* has a strong association with many systemic diseases, including IgAN [14,15,16,17,18]. Periodontal bacteria are also reported to have a strong relationship with systemic diseases [19,20]. Periodontal bacteria are basically anaerobic, because pockets around teeth are anaerobic. Culture methods basically did not work for anaerobic bacteria; therefore, it was difficult to reveal the relationship between periodontal bacteria and many diseases, including IgAN. 

Recently, *Campylobacter rectus* and *Treponema denticola* were identified as the periodontal bacteria related to IgAN [21], using Denaturant gradient gel electrophoresis methods, which did not require bacteria culture [22]. Moreover, *C. recuts* was identified as periodontal bacteria related to the level of proteinuria in IgAN patients [16]. *T. denticola* is one of the red-complex periodontal bacteria which have the most strong pathogenicity of periodontal disease. The red complex of periodontal bacteria consists of *T. denticola,*
*Porphyromonas gingivalis* and *Tannerella forsythia* [23,24]. In this point, the relationship between the red complex of periodontal bacteria and IgAN remained unknown.

In this study, the red complex of periodontal bacteria in tonsils was evaluated in IgAN patients and habitual tonsillitis patients. Moreover, the pathogenicity of IgAN caused by the periodontal bacteria specific to IgAN was confirmed by a mouse model treated with the periodontal bacteria.

## 2. Results

### 2.1. Clinical Characteristics of the Subjects

Tonsil specimens were collected from 23 IgAN patients and 63 habitual tonsillitis patients who underwent tonsillectomy. The average age in the tonsillitis group was lower than in the IgAN group, with statistical significance; however, the differences were not clinically meaningful (Table 1). The average height was similar between the two groups, while body weight in habitual tonsillitis patients was higher than in IgAN patients. 

### 2.2. Detection of Periodontopathic Bacterial Species in Tonsil Specimens

We used PCR to identify three strong-pathogenic periodontal bacterial species (*P. gingivalis*, *T. denticola* and *T. forsythia*, known as the red complex; and *C. rectus*, which is reported to have a relationship with IgAN in tonsil specimens, with species-specific sets of primers (Table 2).

*T. denticola* was rarely detected in tonsil specimens both in IgAN patients and habitual tonsillitis patients (Figure 1). *C. rectus* was detected both in IgAN patients and habitual tonsillitis patients, with no significant different prevalence (Figure 1). *P. gingivalis* in tonsils was detected with significantly higher prevalence in IgAN patients than in habitual tonsillitis patients (*p* < 0.001) (Figure 1). *T. forsythia* was also significantly detected in IgAN patients higher than in the control (*p* < 0.05) (Figure 1). 

To evaluate the effect of *P. gingivalis* and *T. forsythia* on the disease activity of IgA, the IgA patients were divided into two groups; positive or negative of *P. gingivalis* or *T. forsythia*. There were no significant differences between IgAN patients with *P. gingivalis* and those without *P. gingivalis* and between IgAN patients with *T. forsythia* and those without *T. forsythia* regarding proteinuria, hematuria, C3, C4 and serum IgA (Table 3)

To evaluate effect of red complex of periodontal bacteria, the number of red complex periodontal bacteria was compared between IgAN patients and tonsillitis patients. The number of periodontal bacteria in IgAN patients was significantly higher than those of tonsillitis patients (*p* < 0.001) (Figure 2). A total of 48% of IgAN patients had more than one red-complex periodontal bacteria, while 92% of tonsillitis patients were free from those red-complex periodontal bacteria (Figure 2). 

### 2.3. Gd-IgA1 Staining in IgA Patients with or without P. gingivalis, T. forsythia and C. rectus

Galactose-deficient IgA1 (Gd-IgA1) has been reported to be specific to IgAN and have strong relationship with pathogenesis of IgAN [29,30]. The strength of Gd-IgA1 staining was reported to have a relationship with the bacteria in tonsils [31]. Basically, all kidney samples showed positive staining of Gd-IgA1, because these patients were diagnosed as IgA nephropathy. When the staining of Gd-IgA1 in glomerulus more than (+) was treated as positive, IgAN patients with *P. gingivalis* in tonsils tended to be higher than those without *P. gingivalis* in tonsils without statistical significance (Figure 3). There were no significant changes of strength of staining between those with and without *T. forsythia* or *C. rectus*.

### 2.4. IgAN Model Mice Induced by Nasal Administration of P. gingivalis

Among red complex of periodontal bacteria, *P. gingivalis* was chosen for mice treatment experiment, because of higher significant difference between IgAN patients and tonsillitis patients, and because higher prevalence in IgAN patients. *P. gingivalis* or saline (control) was treated into mice once a week from 8 to 16 weeks after birth. At 14, 28, 42 and 56 days after start of administration of *P. gingivalis* blood, and urine and kidney samples were obtained. During the study period, body weight, kidney function, proteinuria had not changed significantly (Appendix A).

### 2.5. Histopathological Analyses of Kidney Tissue of IgAN Model Mice

Histopathological analyses of PAS staining sections revealed that prominent proliferation of mesangial cells and mesangial matrix in the *P. gingivalis* treated group. The mesangial proliferation scores of *P. gingivalis* treated group were significantly greater than those of control group (*p* < 0.05 at days 28 and 42; *p* < 0.01 at days 14 and 56) (Figure 4). 

### 2.6. Immunohistochemical Analyses of Kidney Tissues of IgAN Model Mice

Immunohistochemical analyses using IgA-specific antibodies demonstrated the positive reaction for mesangial region prominently in the mice treated with *P. gingivalis*. The strength of IgA staining was evaluated from (−) to (2+). Typical images of IgA immuno-fluorescence staining are shown in Figure 5. 

When an IgA staining level more than (+) was treated as positive, the positive rate of IgA in the *P. gingivalis*–treated group was significantly greater than in the control groups (Figure 6) (*p* < 0.001 at days 42 and 56). A total of 80% of mice treated with *P. gingivalis* had IgA deposition in the kidney, while only 10% of control mice had IgA deposition at days 42 and 56. 

### 2.7. Scanning Electron Microscopic of Immune Complex

Scanning-electron-microscopic observation revealed that high-density EDD was widely distributed in the mesangial region in the mice kidneys treated with *P. gingivalis* (Figure 7). 

## 3. Discussion

In this study, we revealed that, among the bacteria in the red complex of periodontal bacteria, *P. gingivalis* and *T. forsythia* had a prevalence in the tonsils of IgAN patients that was significantly higher than in those with habitual tonsillitis. A total of 48% of IgAN patients had more than one red-complex periodontal bacteria, while 92% of tonsillitis patients were free from those red-complex periodontal bacteria. Gd-IgA1 staining in the kidney biopsy of IgAN patients with *P. gingivalis* tended to be stronger than those of IgAN without *P. gingivalis*, while there was no significant difference of IgAN patients’ characteristics with or without *P. gingivalis*. Intra-nasal treatment of *P. gingivalis* to mice made the significant mesangial proliferation, and IgA deposition in their kidneys was also confirmed by electron microscopy as dense deposits. These findings indicate that *P. gingivalis* had the pathogenicity of IgAN.

Periodontal bacteria had been reported to have strong relationship with many kinds of systemic diseases. *C. rectus* and *T. denticola* in tonsils had already reported to have strong relationship with IgAN and the clinical remission rate after tonsillectomy [21]. Among periodontal bacteria, red complex has most strong pathogenicity of periodontal diseases. Our results revealed *P. gingivalis* among red complex of periodontal bacteria is specific to IgAN, because anaerobic circumstances in tonsillar crypts may allow these aerobic periodontal bacteria alive in tonsils and *P. gingivalis* may have high ability to colonize in tonsil. Moreover, Gd-IgA staining in IgAN patients with *P. gingivalis* tended to be stronger than those in IgAN patients without *P. gingivalis*, indicating that *P. gingivalis* infection caused Gd-IgA production. These results suggested that *P. gingivalis* infection in tonsils might cause IgAN through Gd-IgA production [25,29,30,31]. 

Nasal administration of *P. gingivalis* to mice caused mesangial proliferation and IgA deposition, which strongly indicated that *P. gingivalis* had pathogenicity of IgAN. Recently, it was reported that nasal-associated lymphoid tissue is the major induction site for nephritogenic IgA in murine IgA nephropathy (ddY mice) [26], indicating that *P. gingivalis* infection in tonsils in humans might work as the major induction site in IgAN patients because murine nasal-associated lymphoid tissue can work as lymphoid reaction in human tonsil. *P. gingivalis,* usually located in periodontal areas as one of red complex of periodontal bacteria, could translocate to tonsils, because both of them were anaerobiotic environment [27], which might cause pathogenesis of IgA nephropathy. Moreover, ddY mice were well-known to be model mice for IgAN, which was genetically similar to human IgAN patients [28,32]. However, in this point there was no IgAN model mice whose characteristics are similar to IgAN in the meaning of the response to infection. In the future, this model might be useful for revealing the mechanism of pathogenesis of IgAN and for finding the new therapeutic targets. 

In this study, there were several limitations. First, we could not evaluate the type of cilia of *P. gingivalis.* Cilia were considered to be related with pathogenicity. This problem should be revealed in the future. Second, the number of samples were limited, while the important results have statistical significance. Third, hematuria in model mice could not be evaluated, because the amounts of urinary samples of mice were limited. Forth, there were many periodontal bacteria which could not be evaluated, while periodontal bacteria whose pathogenicity of periodontal diseases were strong had been checked in this study. Fifth, IgAN had been considered to be caused by multifactorial factors, such as genetic factors and environmental factors. The other causes of IgAN and the relationship between these other factors and *P. gingivalis* infection should be revealed, although *P. gingivalis* might be one of important environmental factors. Sixth, in this study, the tonsils in the patients with habitual tonsillitis were used as control. Although, both of them were free from acute tonsillitis, in order to decrease the complications of tonsillectomy, there was possibility that there were other specific differences of bacterial flora of tonsils between in patients with IgAN and in general populations, although it was ethically impossible to obtain tonsils from general populations. Seventh, there were many bacteria in tonsils, including periodontal bacteria. This study did not indicate that there was no pathogenicity of bacteria, except *P. gingivalis.* Moreover, IgAN patients with double bacterial infections were observed in our results, and this might indicate some possibility that double infections might have more pathogenesis than single infection. These problems should be confirmed in a further study.

In conclusion, the prevalence of *P. gingivalis*, one of red complex of periodontal bacteria, was significantly higher in IgAN patients than in habitual tonsillitis patients. Nasal administration of *P. gingivalis* in mice caused mesangial proliferation and IgA deposition, suggesting that *P. gingivalis* has pathogenesis of IgAN.

## 4. Materials and Methods

### 4.1. Ethics Statement

This study was conducted in full adherence to the Declaration of Helsinki. The study protocol was approved by the Ethics Committee of Hyogo College of Medicine (Approval Number: 202004-211, 15 Mar 2013) and Osaka University Graduate School of Dentistry (Approval Number: H29-E9, 15 June 2017). Prior to specimen collection, all subjects were informed of the study protocol, and they provided written informed consent. These approvals were extended every three years until now.

### 4.2. Subjects and Specimens

Twenty-three patients (9 men and 14 women; average age, 27.3 (22–31) years old) were diagnosed as IgA nephropathy in Hyogo College of Medicine Hospital by kidney biopsy. These patients were enrolled from November 2017 to March 2020, when they received tonsillectomy in Meiwa Hospital. Sixty-three habitual tonsillitis patients (26 men and 37 women; average age = 36.2 (18–50) years old) also enrolled when they received tonsillectomy in Meiwa Hospital. Habitual tonsillitis is defined by the patients who get acute tonsillitis more than 4 times a year or more than 5 or 6 times in two years. During tonsillectomy periods, both patients with IgAN and patients with habitual tonsillitis were free from acute tonsillitis, because acute tonsillitis increases the complications of tonsillectomy. One of the two tonsils removed by tonsillectomy was used for the following cDNA extraction.

### 4.3. DNA Extraction

Bacterial cDNA was extracted from tonsil specimens, using a method described previously [21]. Briefly, total RNA was extracted from homogenized tonsil specimens using TRIzol (Invitrogen, Carlsbad, CA, USA). Then, cDNA was synthesized from 0.4 mg of RNA by a reverse-transcription reaction, using SuperScript II Reverse Transcriptase (Invitrogen). 

### 4.4. PCR Detection of the Periodontopathic Bacterial Species

PCR was performed by using bacterial DNA extracted from the tonsil specimens to detect 4 periodontopathic bacterial species (*P. gingivalis*, *T. denticola*, *T. forsythia* and *C. rectus*) with species-specific sets of primers and TaKaRa Ex Taq polymerase (Takara Bio. Inc., Otsu, Japan; Table 2) [33,34,35,36]. PCR amplification was performed with the following cycling parameters: initial denaturation at 95 °C for 4 min; followed by 30 cycles at 95 °C for 30 s, 58 °C for 30 s and 72 °C for 30 s; and with a final extension at 72°C for 7 m. The PCR products were fractionated in a 1.5% (*w/v*) agarose gel-Tris-acetate-EDTA buffer, then stained with ethidium bromide (0.5 μg/mL) and visualized under UV light. 

### 4.5. Gd-IgA1 Staining for Kidney Biopsy Samples in IgAN Patients

Gd-IgA1 staining has been reported to be useful for diagnosis of IgAN [37], and the strength of its staining was reported to have relationship with the bacteria in tonsils [31]. IF staining for Gd-IgA1 in kidney tissues from IgAN patients was performed by using anti-human Gd-IgA1 (KM55) Rat IgG (TECAN, Mannedorf, Switzerland), a Gd-IgA1- specific monoclonal antibody that has specific immunoreactivity for glomerular IgA deposition and can thus reflect the pathogenesis of IgAN [38]. The Alexa Fluor 488-labeled donkey anti-rabbit IgG (Abcam, Cambridge, UK) secondary antibody was used for detection. Based on the degree of staining, IgAN patients were divided into two groups: (i) strong (1+) and (ii) weak (±) (there was no patient of negative (−) staining).

### 4.6. P. gingivalis Strains and Culture Medium

*P. gingivalis* strain ATCC 33277 (fimA type I) was selected from the stock culture collection in our laboratory [39]. *P. gingivalis* strain was grown anaerobically at 37 °C for 24 h in trypticase soy broth supplemented with yeast extract (1 mg/mL), hemin (5 μg/mL) and menadione (1 μg/mL), as previously described [40].

### 4.7. IgA Model Mice Induced by Nasal Administration of P. gingivalis

All mice were treated humanely in accordance with National Institutes of Health and AERI-BBRI Animal Care and Use Committee guidelines. The study using the mice IgA nephropathy model induced by *P. gingivalis* was approved by the Committee on Animal Experiments of the Hyogo College of Medicine (approval number: 19-018, 25 June 2019). *P. gingivalis* was used as disease model, and normal saline was used as control. *P. gingivalis* was treated via nasal administration, because mice do not have tonsils, and nasal-associated lymphoid tissue in mice could work similarly to tonsils in human [26]. *P. gingivalis* or saline continued to be treated once a week via nasal administration, under anesthetization, from 8 weeks after birth until end of the experiments. Overnight-cultured *P. gingivalis* strains were adjusted to OD600 = 1.0 (equivalent to 1 × 10^9^ CFU/mL), using PBS. Then, the concentration of these bacterial solutions to 10 and 100 times were performed, which were equivalent to 1 × 10^10^ to 1 × 10^11^ CFU/mL, respectively. Kidney samples and urine samples were obtained at days 14, 28, 42 and 56 after the administration of bacteria or saline. 

### 4.8. Analysis of Urinary and Serum Components

At 14, 28, 42 and 56 days after treatment of bacteria, mice were transferred into a metabolic container and the excreted urine was collected by pipette into a sterile tube. Under inhalation anesthetic with isoflurane (Pfizer Japan Inc., Tokyo, Japan), blood and kidney samples were collected, and serum was collected by centrifugation at 3000 rpm for 10 minutes at 4 °C. Urinary levels of albumin were measured by using a mouse albumin ELISA kit, and creatinine (EXOCELL Division of Ethos Biosciences, Inc., Newtown Square, PA, USA) and urinary creatinine (U-CRE) levels were measured by Creatinine (urinary) Colorimetric Assay Kit (Cayman chemical, Ann Arbor, MI, USA). Serum levels of creatinine (CRE) and blood urea nitrogen (BUN) were measured by using i-STAT (Abbott, Princeton, NJ, USA).

### 4.9. Histological Evaluation of the Kidney

The excised kidney tissue was fixed with 3.7% formaldehyde (FUJIFILM Wako Pure Chemical Co., Osaka, Japan) diluted in PBS and then embedded in paraffin to prepare 3 μm tissue sections. Tissue sections were subjected to Periodic Acid-Schiff (PAS) staining according to the method of Schaart et al. [41] and observed under an optical microscope (BX53F, OLYMPUS, Tokyo, Japan) to assess the proliferation of glomerular mesangial cells and mesangial matrix in glomeruli. Then, mesangial proliferation scores were calculated based on the percentage of glomeruli with mesangial cells’ proliferation and increased mesangial matrix among 50 glomeruli in PAS-stained sections [42,43]. Additionally, immune-fluorescence staining of IgA in the kidney tissues was detected by using standard immunohistochemical techniques with IgA-specific antibodies. The primary antibody was Purified Rat Anti-Mouse IgA (BD Biosciences, San nose, CA, USA). Secondary antibodies were Goat Anti-Rat IgG (Alexa Fluor-488) (Invitrogen, Carlsbad, CA, USA). The blocking procedure was performed by using Normal Donkey Serum (Jackson ImmunoResearch, West Grove, PA, USA). Fluorescence immunostaining was performed by using these antibodies. Stained sections were observed under an all-in-one fluorescence microscope (BZ-X700; Keyence, Osaka, Japan). 

### 4.10. Transmission Electron Microscopy

For pre-fixation, excised kidney-tissue specimens were immersed in a solution of 2% glutaraldehyde and 2% paraformaldehyde in PBS (0.1 M, pH 7.4) for 16–18 h. Then, post-fixation was performed in 2% osmium tetroxide for 1.5 h. After washing with PBS, the specimens were dehydrated in a graded ethanol series and embedded in low-viscosity resin (Spurr resin; Polysciences, Warrington, PA, USA). Then, 80 nm–ultrathin sections were prepared by using an ultramicrotome (EM-UC 7; Leica, Tokyo, Japan) and stained with uranyl acetate and lead citrate. Specimens were observed under a transmission electron microscope (H-7650; HITACHI, Tokyo, Japan).

### 4.11. Statistical Analysis

Statistical analyses were performed by using SATA15 (StataCorp LLC, College Station, TX, USA. Comparisons between two groups were performed by using the Chi-square test. A *p* < 0.05 was considered to be significant.

## Figures and Tables

**Figure 1 ijms-22-13022-f001:**
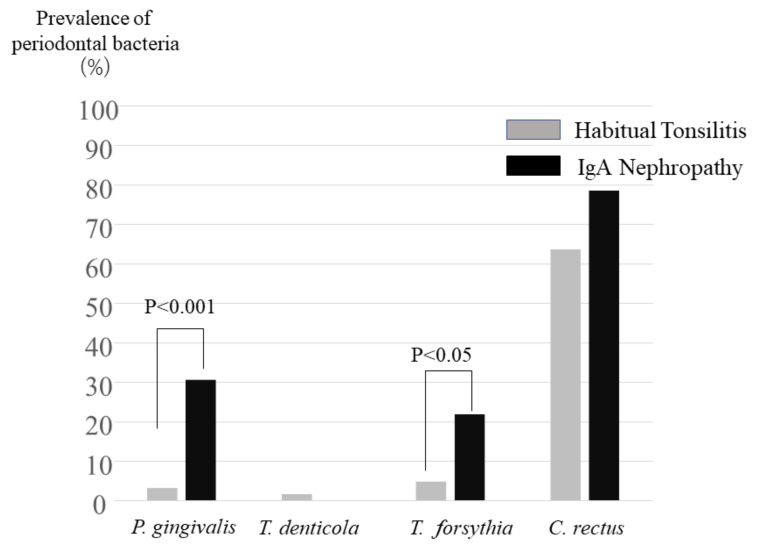
Prevalence of periodontal bacteria. Prevalence of red complex of periodontal bacteria (*P. gingivalis*, *T. denticola* and *T. forsythia*) and *C. rectus* in tonsils with IgA nephropathy and tonsillitis patients were shown. Prevalence of *P. gingivalis* in IgA nephropathy patients was significantly higher than that in tonsillitis (*p* < 0.001). Prevalence of *T. forsythia* in IgA nephropathy patients was also significantly higher than that in tonsillitis (*p* < 0.05).

**Figure 2 ijms-22-13022-f002:**
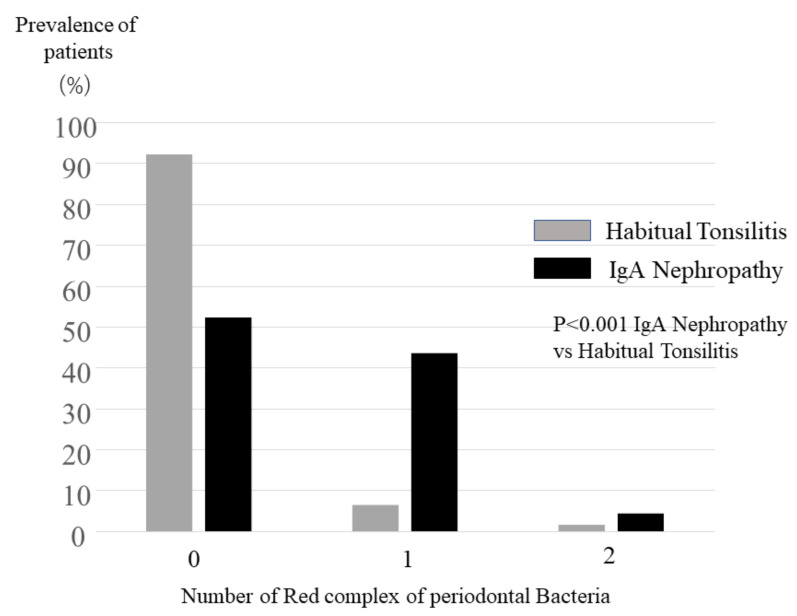
Prevalence of numbers of periodontal bacteria out of three red complex of bacteria. Overall, 92.0% of tonsillitis patients were free from red complex of periodontal bacteria, while 52.1% of tonsillitis patients were free from those bacteria; 43.4% of IgA nephropathy patients were suffering from one periodontal bacterium, while only 6.3% of tonsillitis patents had one from red complex of periodontal bacteria. The distribution of the number of periodontal bacteria in IgA nephropathy patients was significantly different to that in tonsillitis patients (*p* < 0.001).

**Figure 3 ijms-22-13022-f003:**
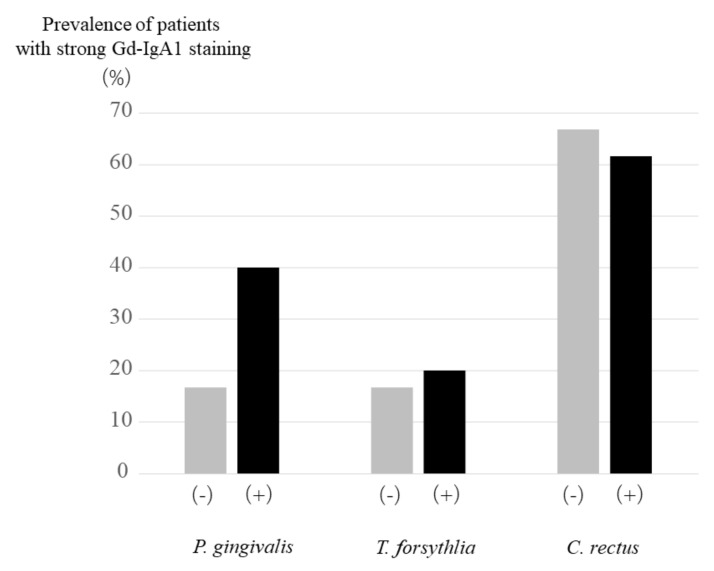
Prevalence of IgA patients with strong galactose-deficient IgA1 (Gd-IgA1) staining in the kidney biopsy specimen. Prevalence of IgA patients with strong Gd-IaA1 staining in the kidney biopsy specimen in IgA patients with *P. gingivalis* tended to be higher than that in IgA patients without *P. gingivalis*.

**Figure 4 ijms-22-13022-f004:**
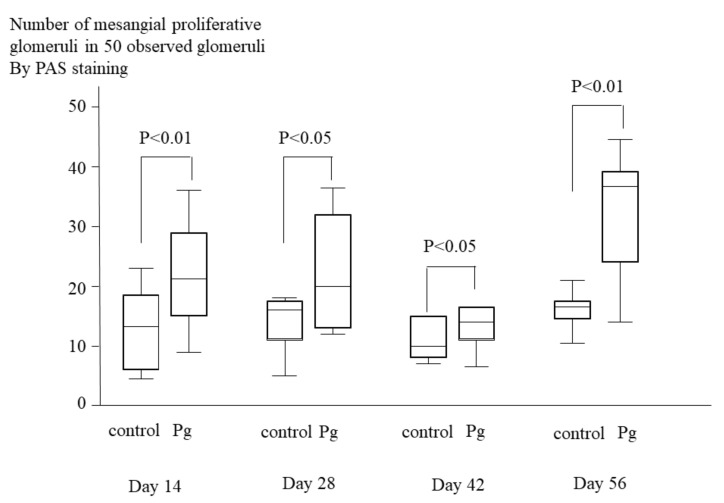
Numbers of mesangial proliferative glomeruli in 50 observed glomeruli in mice treated with *P. gingivalis* or saline (control). Glomeruli were evaluated by PAS staining. Each group consists of 10 mice. Numbers of mesangial proliferative glomeruli in 50 observed glomeruli in mice treated with *P. gingivalis* were significantly higher than those in control mice at day 14 (*p* < 0.01), at day 28 (*p* < 0.05), at day 42 (*p* < 0.05) and at day 56 (*p* < 0.01) after treatment of bacteria.

**Figure 5 ijms-22-13022-f005:**
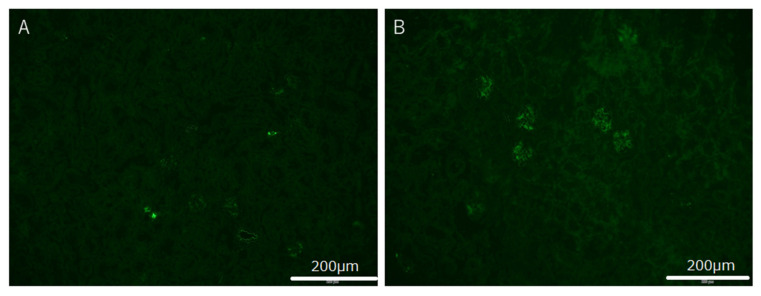
Typical image of IgA immune-fluorescence staining in mice IgA nephropathy model treated with *P. gingivalis*. (**A**) None of the IgA immune-fluorescence stainings in control mice was observed. (**B**) Positive immune-fluorescence staining was observed in mice treated with *P. gingivalis* at day 56 after beginning treatment. IgA staining was observed in mesangial area in glomeruli.

**Figure 6 ijms-22-13022-f006:**
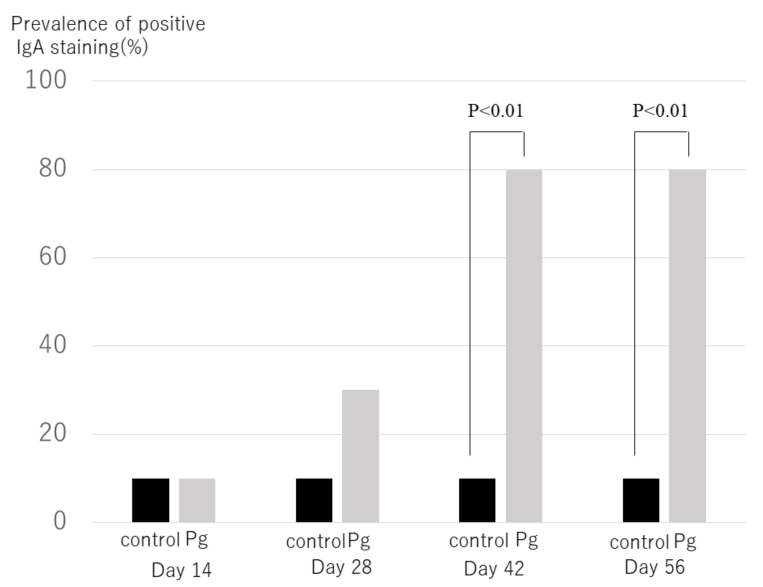
Prevalence of mice with positive IgA immune-fluorescence staining. In each mouse treated with *P. gingivalis* or saline (control) via nasal administration, IgA immune-fluorescence staining in glomeruli was observed under same condition. Each group consists of 10 mice. Numbers of mice with IgA immune-fluorescence staining treated with *P. gingivalis* were significantly higher than those of control mice at day 42 (*p* < 0.01), at day 42 and at day 56 (*p* < 0.01) after treatment of bacteria.

**Figure 7 ijms-22-13022-f007:**
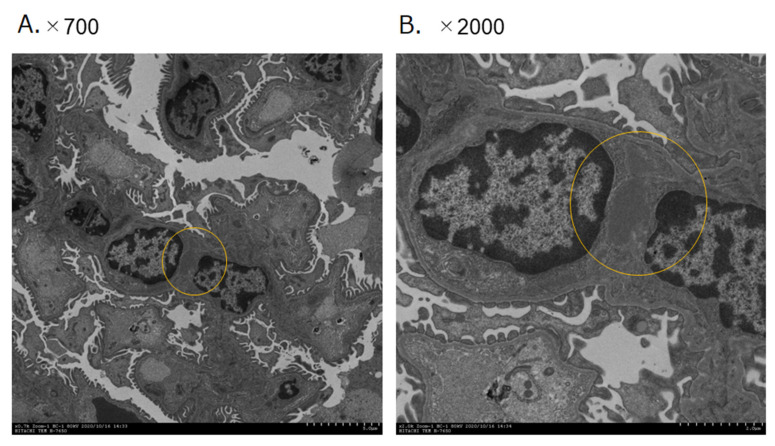
Typical images of immune deposit in mesangial area of mice glomeruli treated with *P. gingivalis* at day 56 after treatment of bacteria, as observed by transmission electron microscopy. Immune deposit in mesangial area of mice glomeruli treated with *P. gingivalis* at day 56 after treatment of bacteria is indicated by yellow circle: (**A**) ×700 magnification and (**B**) ×2000 magnification.

**Table 1 ijms-22-13022-t001:** Patients characteristics.

	IgAnephropathy		Habitual Tonsilitis		*p*
Age	33 ± 14		27 ± 7		<0.01
Sex	9 Male	14 Female	26 Male	37 Female	NS
Height (cm)	160 ± 10		162 ± 20		NS
Weight (Kg)	54 ± 11		62 ± 15		<0.05
Proteinuria (g/gCre)	0.9 ± 1.1		N/A		N/A

N/A; Not available. NS; Not significant.

**Table 2 ijms-22-13022-t002:** Primers for detection of bacteria-related periodontal diseases.

Purpose	Sequence (5′-3′)	Size (bp)	Ref.
**Universal primer**			
**(positive control)**			
PA	AGA GTT TGA TCC TGG CTC AG	315	[25]
PD	GTA TTA CCG CGG CTG CTG		
**Detection of periodontitis-related species**			
*Porphyromonas gingivalis*	CCG CAT ACA CTT GTA TTA TTG CAT GAT A	267	[26]
	AAG AAG TTT ACA ATC CTT AGG ACT GTC T		
*Treponema denticola*	AAG GCG GTA GAG CCG CTC A	311	[27]
	AGC CGC TGT CGA AAA GCC CA		
*Tannerella forsythia*	GCG TAT GTA ACC TGC CCG CA	641	[28]
	TGC TTC AGT GTC AGT TAT ACC T		
*Campylobacter rectus*	TTT CGG AGC GTA AAC TCC TTT TC	598	[28]
	TTT CTG CAA GCA GAC ACT CTT		

**Table 3 ijms-22-13022-t003:** Patient backgrounds with or without *P. gingivalis* and *T. forsythia*.

	*P. gingivaris*		
	(−) (N = 16)	(+) (N = 7)	*p*
Protenuria (g/acre)	1.1 ± 0.8	1.1+1.4	NS
C3 (mg/dL)	98.5 ± 4.5	100.7 ± 8.4	NS
C4 (mg/dL)	25.2 ± 1.6	22.7 ± 3.0	NS
IgA (mg/dL)	368 ± 49	398 ± 36	NS
	*T. forsythia*		
	(−) (N = 18)	(+) (N = 5)	*p*
Protenuria (g/gcre)	1.1 ± 0.3	1.1 ± 0.1	NS
C3 (mg/dL)	98.5 ± 4.5	99.5 ± 4.7	NS
C4 (mg/dL)	23.4 ± 1.6	30.0 ± 6.0	NS
IgA (mg/dL)	380 ± 33	384 ± 148	NS

NS: Not significant.

## Data Availability

The data presented in this study are available upon request from the corresponding author. The data are not publicly available, due to ethical and privacy limitations.

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
