# Peer review of "Relationship between IgA Nephropathy and Porphyromonas gingivalis; Red Complex of Periodontopathic Bacterial Species"

_ijms, 2021, doi:10.3390/ijms222313022_

Round 1

Reviewer 1 Report

  1. Why did the authors only compare IgA patients to chronic tonsillitis patients? The reviewer thinks that the authors need to consider the comparison group.
  2. Can it be argued that the mere abundance of P. gingivalis in the tonsils of patients with IgA caused IgA nephropathy?
  3. Many studies have reported the association between tonsil and IgA, but can it be said that IgA nephropathy was developed by invasion without infection? Investigating the microbiology of tonsil in IgA patients, the microbiology of tonsillitis, and the microbiology of tonsil in the general population needs. Although there is no pathogen in the tonsil of the general population, it is necessary to prove that the microbiology of patients with IgA and tonsillitis is the same.
  4. In addition, the microbiology of tonsillitis in patients without IgA versus the development of tonsillitis in patients with IgA should also be compared.
  5. In the in vivo study, only P. gingivalis was administered to mice, but it is thought that P. gingivalis can be explained as a pathogen that causes IgA only when it is confirmed that IgA is not expressed when other pathogens are administered. This is because, in humans, not all patients with IgA have P. gingivalis, and since this is a normal oral flora, tonsillitis does not mean that all patients develop IgA nephropathy. It is necessary to distinguish whether IgA is expressed when a pathogen other than P. gingivalis is administered, or whether IgA is expressed due to the infection itself.
  6. In Table 3, C3, C4, and IgA levels are not significant. What was the basis for dividing Hematuria into grades? The level of occult blood is meaningless because it is not a quantitative test, and hematuria does not indicate the severity of the disease.
  7. In figure 3 and figure 6, there are no bar graphs for patients with IgA and tonsilitis.

Author Response

Thank you very much for useful and instructive comments. According to your comments we revised manuscript. We hope this revised manuscript may reply for your requirements.

REVIEWER 1

1.Why did the authors only compare IgA patients to chronic tonsillitis patients? The reviewer thinks that the authors need to consider the comparison group.

Answer

Thank you for your good suggestions. We used “chronic tonsilitis patients” as the control group name. This might make mis-understanding. The exact name is “habitual tonsilitis patients”. The definition of habitual tonsilitis is the patients who get acute tonsilitis more than 4 times a year or, more than 5-6 times in two years. During the tonsillectomy periods, both habitual tonsilitis patients and IgA nephropathy patients were free from acute tonsilitis, because acute tonsilitis increase the complication of tonsilitis. We had tried to enroll all patients with tonsillectomy. Only three patients were suffering from pathogenic tonsillitis maybe plantar pustulosis, whose detail disease record could not be obtained, because our ethical approvement did not include this process in the patients without IgA nephropathy. We excluded these three patients from control. There was no tonsillectomy because of other reasons, such as sleep anemia syndrome. We changed the name of control from chronic tonsilitis to habitual tonsilitis, and we added explanation as shown below in patients enrolment section.

In methods section

The definition of habitual tonsilitis is the patients who get acute tonsilitis more than 4 times a year or, more than 5-6 times in two years.

During tonsillectomy periods, both patients with IgAN and patients with habitual tonsilitis were free from acute tonsilitis, because acute tonsilitis increases the complications of tonsillectomy.

  1. Can it be argued that the mere abundance of P. gingivalis in the tonsils of patients with IgA caused IgA nephropathy?

Thank you for your important caution. IgAN might be caused by multifactorial factors, such as genetic factors and environmental factors. In these points, we suspected that among environmental factors, P. gingivalis infections might be important, but there were many other important factors. Therefore we add following paragraph in the limitation in the discussion sections.

Fifth, IgAN had been considered to be caused by multifactorial factors, such as genetic factors, and environmental factors. The other causes of IgAN and the relationship between these other factors and P. gingivalis infection should be revealed, although P. gingivalis might be one of important environmental factors.

  1. Many studies have reported the association between tonsil and IgA, but can it be said that IgA nephropathy was developed by invasion without infection? Investigating the microbiology of tonsil in IgA patients, the microbiology of tonsillitis, and the microbiology of tonsil in the general population needs. Although there is no pathogen in the tonsil of the general population, it is necessary to prove that the microbiology of patients with IgA and tonsillitis is the same.

Thank you for your important caution again. As you pointed, we checked the red-complex of periodontal bacteria, and C.rectus. Therefore, our data could not conclude there were differences of whole microbiology in tonsils with IgA patients and habitual tonsilitis. As you mentioned, it is very interesting and important points whether there is the difference between the microbiology between the tonsils with IgAN patients and those with general populations. But, it was ethically impossible to obtain tonsils from general populations. Periodontal bacteria was basically anaerobic, therefore swab method could not detect these bacteria, because swab can get the sample only from surface of tonsil rather than from deep crypts in tonsils. But, your caution is important, therefore we added the following paragraph in the limitations.

Sixth, in this study, the tonsils in the patients with habitual tonsilitis were used as control. Although, both of them were free from acute tonsilitis, in order to decrease the complications of tonsillectomy, there was possibility that there were other specific differences of bacterial flora of tonsils between in patients with IgAN and in general populations, although it was ethically impossible to obtain tonsils from general populations.

  • In addition, the microbiology of tonsillitis in patients without IgA versus the development of tonsillitis in patients with IgA should also be compared.

 Thank you for your important and interesting comments. When IgAN patients got acute tonsilitis, they sometimes manifested the macrohematuria. It was a very interesting question that there were some specific features in tonsils with IgAN during acute tonsilitis. But, unfortunately, in order to decrease the complication of tonsillectomy, during acute tonsillitis, tonsillectomy is usually postponed after they get recovery.

  • In the in vivo study, only gingivalis was administered to mice, but it is thought that P. gingivalis can be explained as a pathogen that causes IgA only when it is confirmed that IgA is not expressed when other pathogens are administered. This is because, in humans, not all patients with IgA have P. gingivalis, and since this is a normal oral flora, tonsillitis does not mean that all patients develop IgA nephropathy. It is necessary to distinguish whether IgA is expressed when a pathogen other than P. gingivalis is administered, or whether IgA is expressed due to the infection itself.

Thank you for your very important comments. Preliminarily, we had already tried to confirm the pathogenicity of the periodontal bacteria using C.rectus, and T. denticola. But, basically there was no IgA deposition in mesangial areas of glomeruli. We hope these results might support the pathogenicity of P. gingivalis. However, negative data was difficult to make conclusion, because some technical problems can make negative data. This article was very important to provide basic structure to confirm the pathogenicity of bacteria, because our data included positive results. In the future, we should confirm the pathogenicity of bacteria using other bacteria or double infection of bacteria. But, we added following sentences in limitation, because your comments are important.

Seventh, there were many bacteria in tonsils including periodontal bacteria. This article did not indicate there was no pathogenicity of bacteria except P. gingivalis. Moreover, IgAN patients with double bacterial infections were observed in our results, which might indicate some possibility that double infections might have more pathogenesis than single infection. These problems should be confirmed in further study.

  • In Table 3, C3, C4, and IgA levels are not significant. What was the basis for dividing Hematuria into grades? The level of occult blood is meaningless because it is not a quantitative test, and hematuria does not indicate the severity of the disease.

 Thank you for important points. As you points, we eliminated the grade of hematuria from table1, 3.

  • In figure 3 and figure 6, there are no bar graphs for patients with IgA and tonsilitis.

Thank you for your good points. These figures means that prevalence of positive staining. IgA and Gd-IgA staining can provide positive or negative of staining.

Reviewer 2 Report

The authors revealed the relationship between IgAN and periodontopathic bacterial species. Interesting idea, might be used in general practice and should be verified in larger cohort. Statistical analysis, figures, tables and the structure of the article were well prepared. I agree with publication in current form.

Author Response

Thank you very much for useful and instructive comments. According to your comments we revised manuscript. We hope this revised manuscript may reply for your requirements.

REVIWER2

The authors revealed the relationship between IgAN and periodontopathic bacterial species. Interesting idea, might be used in general practice and should be verified in larger cohort. Statistical analysis, figures, tables and the structure of the article were well prepared. I agree with publication in current form.

Thank you very much for warm comments.

Round 2

Reviewer 1 Report

I think that the authors have adequately corrected their paper according to the reviewer’s recommendations overall. I agree with the limitations of the revision from the authors.